# Variational Flow Matching for Graph Generation

**Floor Eijkelboom**[*]
UvA-Bosch Delta Lab
University of Amsterdam

**Grigory Bartosh**[*]
AMLab
University of Amsterdam

**Christian A. Naesseth**
UvA-Bosch Delta Lab
University of Amsterdam

**Max Welling**
UvA-Bosch Delta Lab
University of Amsterdam

**Jan-Willem van de Meent**
UvA-Bosch Delta Lab
University of Amsterdam

## Abstract

We present a formulation of flow matching as variational inference, which we refer to as variational flow matching (VFM). Based on this formulation we develop CatFlow, a flow matching method for categorical data. CatFlow is easy to implement, computationally efficient, and achieves strong results on graph generation tasks. The key observation in VFM is that we can parameterize the vector field of the flow in terms of a variational approximation of the posterior probability path, which is the distribution over possible end points of a trajectory. We show that this variational interpretation admits both the CatFlow objective and the original flow matching objective as special cases. We also relate VFM to score-based models, in which the dynamics are stochastic rather than deterministic, and derive a bound on the model likelihood based on a reweighted VFM objective. We evaluate CatFlow on one abstract graph generation task and two molecular generation tasks. In all cases, CatFlow exceeds or matches performance of the current state-of-the-art.

## 1 Introduction

In recent years, the field of generative modeling has seen notable advancements. In image generation [41, 45], the development and refinement of diffusion-based approaches — specifically those using denoising score matching [59] — have proven effective for generation at scale [15, 54]. However, while training can be done effectively, the constrained space of sampling probability paths in a diffusion requires tailored techniques to work [52, 64]. This is in contrast to more flexible approaches such as continuous normalizing flows (CNFs) [8], that are able to learn a more general set of probability paths than diffusion models [53], at the expense of being expensive to train as they require one to solve an ODE during each training step (see e.g. [5, 46, 14]).

Recently, Lipman and collaborators [27] proposed flow matching (FM), an efficient and simulation-free approach to training CNFs. Concretely, they use a per-sample interpolation to derive a simpler objective for learning the marginal vector field that generates a desired probability path in a CNF. This formulation provides equivalent gradients without explicit knowledge of the (generally intractable) marginal vector field. This work has been extended to different geometries [7, 23] and various applications [60, 9, 13, 24]. Similar work has been proposed concurrently in [32, 1].

This paper identifies a reformulation of flow matching that we refer to as variational flow matching (VFM). In flow matching, the vector field at any point can be understood as the expected continuation toward the data distribution. In VFM, we explicitly parameterize the learned vector field as an expectation relative to a variational distribution. The objective of VFM is then to minimize the

---

[*]These authors contributed equally to this work.

38th Conference on Neural Information Processing Systems (NeurIPS 2024).

Kullback-Leibler (KL) divergence between the posterior probability path, i.e. the distribution over possible end points (continuations) at a particular point in the space, and the variational approximation.

We show that VFM recovers the original flow matching objective when the variational approximation is Gaussian and the conditional vector field is linear in the data. Under the assumption of linearity, a solution to the VFM problem is also exact whenever the variational approximation matches the *marginals* of the posterior probability path, which means that we can employ a fully-factorized variational approximation without loss of generality.

While VFM provides a general formulation, our primary interest in this paper is its application to graph generation, where the data are categorical. This setting leads to a simple method that we refer to as CatFlow, in which the objective reduces to training a classifier over end points on a per-component basis. We apply CatFlow to a set of graph generation tasks, both for abstract graphs [35] and molecular generation [40, 19]. By all metrics, our results match or substantially exceed those obtained by existing methods.

## 2   Background

### 2.1   Transport Framework for Generative Modeling and CNFs

Common generative modeling approaches such as normalizing flows [42, 38] and diffusion models [15, 54] parameterize a transformation $\varphi$ from some initial tractable probability density $p_0$ – typically a standard Gaussian distribution – to the target data density $p_1$. In general, there is a trade-off between allowing $\varphi$ to be expressive enough to model the complex transformation while ensuring that the determinant term is still tractable. One such transformation is a continuous normalizing flow (CNF).

Any time-dependent[1] vector field $v_t : [0, 1] \times \mathbb{R}^D \to \mathbb{R}^D$ gives rise to such a transformation – called a *flow* – as such a field induces a time-dependent diffeomorphism $\varphi_t : [0, 1] \times \mathbb{R}^D \to \mathbb{R}^D$ defined by the following ordinary differential equation (ODE):

$$\frac{d}{dt}\varphi_t(x) = v_t(\varphi_t(x)) \text{ with initial conditions } \varphi_0(x) = x. \tag{1}$$

In CNFs, this vector field is learned using a neural network $v_t^\theta$. Through the change of variables formula $p_t(x)$ can be evaluated (see appendix E) and hence one could try and optimize the empirical divergence between the resulting distribution $p_1$ and target distribution. However, obtaining a gradient sample for the loss requires ones to solve the ODE induced during training, making this approach computationally expensive.

### 2.2   Flow Matching

In flow matching [27], the aim is to regress the underlying vector field of a CNF directly on the interval $t \in [0, 1]$. Flow matching leverages the fact that even though we do not have access to the *actual* underlying vector field – which we denote as $u_t$ – and probability path $p_t$, one can construct a per-example formulation by defining *conditional flows*, i.e. the trajectories towards specific datapoints $x_1$. Concretely, FM sets:

$$u_t(x) = \int u_t(x \mid x_1) \frac{p_t(x \mid x_1) p_{\text{data}}(x_1)}{p_t(x)} \mathrm{d}x_1, \tag{2}$$

where $u_t(x \mid x_1)$ is the conditional trajectory. The most common way to define $u_t(x \mid x_1)$ is as the straight line continuation from $x$ to $x_1$, implying one can obtain samples $x \sim p_t(x \mid x_1)$ simply by interpolating samples $x_0 \sim p_0$ for some $p_0$ and $x_1 \sim p_1$,

$$u_t(x \mid x_1) := \frac{x_1 - x}{1 - t} \implies x = tx_1 + (1 - t)x_0 \text{ is a sample } x \sim p_t(x \mid x_1). \tag{3}$$

Crucially, the flow matching objective

$$\mathcal{L}_{\text{FM}}(\theta) = \mathbb{E}_{t \sim [0,1], x \sim p_t(x)} \left[ \left\| v_t^\theta(x) - u_t(x) \right\|_2^2 \right] \tag{4}$$

---

[1]Time dependence is denoted through the subscript $t$ throughout this paper.

is equivalent in expectation (up to a constant) to the conditional flow matching objective

$$\mathcal{L}_{\text{CFM}}(\theta) = \mathbb{E}_{t \sim [0,1], x_1 \sim p_{\text{data}}(x_1), x \sim p_t(x|x_1)} \left[ \left\| v_t^\theta(x) - u_t(x \mid x_1) \right\|_2^2 \right]. \tag{5}$$

An advantage of flow matching is that this conditional trajectory $u_t(x \mid x_1)$ can be chosen to make the problem tractable. The authors show that diffusion models can be instantiated as flow matching with a specific conditional trajectory, but also show that assuming a simple, straight-line trajectory leads to more efficient training. Note that in contrast to likelihood-based training of CNFs, flow matching is simulation free, leading to a scalable approach to learning CNFs.

## 3 Variational Flow Matching for Graph Generation

We derive CatFlow through a novel, variational view on flow matching we call *Variational Flow Matching (VFM)*. The VFM framework relies on two insights. First, we can define the marginal vector field and its approximation in terms of an expectation with respect to a distribution over end points of the transformation. This implies that we can map a flow matching problem onto a variational counterpart. Second, under typical assumptions on the forward process, we can decompose the expected conditional vector field into components for individual variables which can be computed in terms of the *marginals* of the distribution over end points of the conditional trajectories. This implies that, without loss of generality, we can solve a VFM problem using a fully-factorized variational approximation, providing a tractable approximate vector field. For categorical data, the corresponding vector field can be computed efficiently via direct summation. This results in a closed-form objective to train CNFs for categorical data, which we refer to as *CatFlow*. We develop the theory of VFM in section 3 and we relate VFM to flow matching and score-based diffusion in section 4.

### 3.1 Flow Matching using Variational Inference

In any flow matching problem, the vector field in eq. (2) can be expressed as an expectation

$$u_t(x) = \int u_t(x \mid x_1) p_t(x_1 \mid x) \mathrm{d}x_1 = \mathbb{E}_{p_t(x_1|x)} \left[ u_t(x \mid x_1) \right], \tag{6}$$

where $p_t(x_1 \mid x)$ is the posterior probability path, the distribution over possible end points $x_1$ of paths passing through $x$ at time $t$,

$$p_t(x_1 \mid x) := \frac{p_t(x, x_1)}{p_t(x)}, \qquad\qquad p_t(x, x_1) := p_t(x \mid x_1) \, p_{\text{data}}(x_1). \tag{7}$$

This makes intuitive sense: the velocity in point $x$ is given by all the continuations from $x$ to final points $x_1$, weighted by how likely that final point is given that we are at $x$. Note that to compute $u_t(x)$, one has to evaluate a joint integral over $D$ dimensions.

This observation leads us to propose a change in parameterization of the learned vector field. Rather than predicting the components of the vector field directly, we can define an approximate vector field in terms of an expectation with respect to a variational distribution $q_t^\theta$ with parameters $\theta$,

$$v_t^\theta(x) := \int u_t(x \mid x_1) \, q_t^\theta(x_1 \mid x) \, \mathrm{d}x_1. \tag{8}$$

Clearly, in this construction $v_t^\theta(x)$ will be equal to $u_t(x)$ when $q_t^\theta(x_1 \mid x)$ and $p_t(x_1 \mid x)$ are identical. This implies that we can map a flow matching problem onto a variational inference problem.

Concretely, we can define a variational flow matching problem by minimizing the Kullback-Leibler (KL) divergence from $p_t$ to $q_t^\theta$, which we can express as

$$\mathbb{E}_t \left[ \text{KL}\big(p_t(x)p_t(x_1 \mid x) \,\|\, p_t(x)q_t^\theta(x_1 \mid x)\big) \right] = -\mathbb{E}_{t,x,x_1} \left[ \log q_t^\theta(x_1 \mid x) \right] + \text{const}, \tag{9}$$

where $t \sim \text{Uniform}(0, 1)$ and $x, x_1 \sim p_t(x, x_1)$ (see appendix A.1 for derivations). This leads us to propose the *variational flow matching (VFM)* objective

$$\mathcal{L}_{\text{VFM}}(\theta) = -\mathbb{E}_{t,x,x_1} \left[ \log q_t^\theta(x_1 \mid x) \right]. \tag{10}$$

While this variational formulation of flow matching is promising, two potential drawbacks emerge in practical applications. First, although it is feasible to reformulate any flow matching problem as

a variational inference task, doing so requires learning an approximation of a potentially complex, high-dimensional distribution $p_t(x_1 \mid x)$ (including any correlations between the different $x_1^d$). Second, representing $v_t^\theta(x)$ as an expectation may pose intractability challenges. Interestingly, under typical assumptions about the conditional velocity field in flow matching, this objective simplifies significantly, making it no more computationally demanding than standard flow matching, a point we will further address in section 3.2.

## 3.2 Mean-Field Variational Flow Matching

**Decomposing the conditional vector field.** At first glance, we do not seem to obtain much from this variational view due to the intractability of $p_t(x_1 \mid x)$ and $v_t^\theta(x)$. Fortunately, we can simplify the objective and the calculation of the marginal vector field under the typical case where the conditional vector field $u_t(x \mid x_1)$ is linear in $x_1$, such as in straight line interpolations commonly used in flow matching. This leads to two simplifications. First, we notice that the expected value of a linear conditional velocity field simply equals the conditional velocity field towards the expectation/mean of the distribution we are considering, i.e.

$$u_t(x) = \mathbb{E}_{p_t(x_1 \mid x)}\left[u_t(x \mid x_1)\right] = u_t(x \mid \mathbb{E}_{p_t}\left[x_1 \mid x\right]) \text{ if } u_t(x \mid x_1) \text{ is linear in } x_1. \quad (11)$$

This means that as long as our variational distribution has the same mean as $p_t(x_1 \mid x)$, we will learn the same underlying field. Second, we realize that the expected value of $x_1^d$ only depends on $p_t(x_1^d \mid x)$, and thus as such as long as $q_t^\theta(x_1 \mid x)$ has the same marginal expectations $\mathbb{E}\left[x_1^d \mid x\right]$, we will learn the same field as under $p_t(x_1 \mid x)$. In other words, it suffices to learn a fully-factorized approximation $q_t^\theta(x_1 \mid x)$, there is no need to fully characterize the covariance of the posterior probability path. Thich allows us to reduce a high-dimensional variational problem to a series of low dimension problems.

Formally, the following holds:

**Theorem 1.** Assume that the conditional vector field $u_t(x \mid x_1)$ is linear in $x_1$. Then, for any distribution $r_t(x_1 \mid x)$ such that the marginal distributions coincide with those of $p_t(x_1 \mid x)$, the corresponding expectations of $u_t(x \mid x_1)$ are equal, i.e.

$$\mathbb{E}_{r_t(x_1 \mid x)}\left[u_t(x \mid x_1)\right] = \mathbb{E}_{p_t(x_1 \mid x)}\left[u_t(x \mid x_1)\right]. \quad (12)$$

We provide a proof in appendix A.3.

It follows directly from theorem 1 that *without loss of generality* we can consider the considerably easier task of a fully-factorized approximation

$$q_t^\theta(x_1 \mid x) := \prod_{d=1}^{D} q_t^\theta(x_1^d \mid x). \quad (13)$$

We refer to this special case as *mean-field variational flow matching* (MF-VFM), and the VFM objective reduces to

$$\mathcal{L}_{\text{MF-VFM}}(\theta) = -\mathbb{E}_{t,x,x_1}\left[\log q_t^\theta(x_1 \mid x)\right] = -\mathbb{E}_{t,x,x_1}\left[\sum_{d=1}^{D} \log q_t^\theta(x_1^d \mid x)\right]. \quad (14)$$

Even though we use a factorized model, it is worth emphasizing that *due to the linearity of the conditional field, a mean-field variational distribution can learn the solution exactly*.

**Computing the marginal vector field.** To calculate the vector field $v_t^\theta(x)$, we can simply substitute the factorized distribution $q_t^\theta(x_1 \mid x)$ into eq. (8). However, this still requires an evaluation of an expectation. Fortunately, leveraging the linearity condition significantly simplifies this computation, since as long as we have access to the first moment of one-dimensional distributions $q_t^\theta(x_1^d \mid x)$, we can efficiently calculate $v_t^\theta(x)$ by simply considering the conditional field towards it. Note that for many distributions – e.g. Gaussian – learning its parameters is equivalent to learning its expected value. Note that the training procedure will differ for two distinct distributions – e.g. Gaussian versus Categorical – so the form of the distribution $q_t^\theta(x_1 \mid x)$ remains practically important, a flexibility provided through the variational view on flow matching.

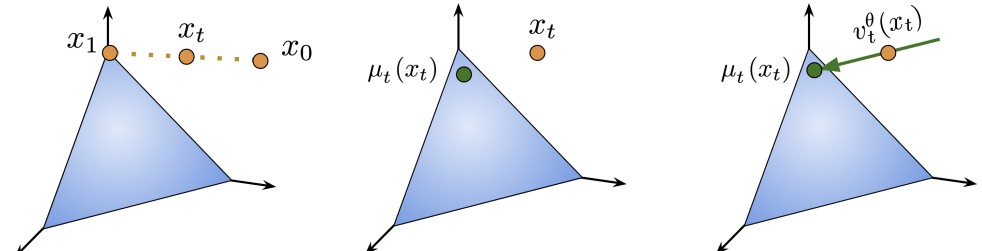

Figure 1: Parameterization of the vector field in CatFlow. Given an interpolant $x_t = tx_1 + (1-t)x_0$, CatFlow predicts a categorical distribution $q_t^\theta(x_1 \mid x_t)$ parameterized by a vector $\mu_t(x_t)$. The resulting construction for the vector field $v_t^\theta(x_t) = (\mu_t(x_t) - x_t)/(1-t)$ ensures that trajectories converge to a point on the simplex at $t = 1$.

If we now use the standard flow matching case of using a conditional vector field based on a linear interpolation, the approximate vector field can be expressed in terms of the first moment of the variational approximation:

$$v_t^\theta(x) = \mathbb{E}_{q_t^\theta(x_1|x)}\left[\frac{x_1 - x}{1-t}\right] = \frac{\mu_1 - x}{1-t}, \qquad \mu_1 := \mathbb{E}_{q_t^\theta(x_1|x)}[x_1]. \qquad (15)$$

Note that this covers both the case of categorical data, which we focus on in this paper, and the case of continuous data, as considered in traditional flow matching methods. We provided the general algorithm for training and generation in appendix B.1.

At first glance, the linearity condition of the conditional vector field $u_t(x \mid x_1)$ in theorem 1 might seem restricting. However, in most state-of-the-art generative modeling techniques, this condition is satisfied, e.g. diffusion-based models, such as flow matching [27, 2, 32], diffusion models [15, 54], and models that combine the injection of Gaussian noise with blurring [44, 16], among others [10, 51].

### 3.3 CatFlow: Mean-Field Variational Flow Matching for Categorical Data

**The CatFlow Objective** In CatFlow, we directly apply the VFM framework to the categorical case. Let our parameterised variational distribution $q_t^\theta(x_1^d \mid x) = \mathsf{Cat}(x_1^d \mid \theta_t^d(x))$, and let us denote the parameters of this categorical as

$$\mu_t^{dk}(x) := q_t^\theta(x_1^d = k \mid x) = \mathbb{E}_{q_t^\theta(x_1|x)}[\mathbb{I}[x_1^d = k]]. \qquad (16)$$

Then, the $d$th component of the learned vector field is

$$v_t^{\theta,d}(x) := \sum_{k=1}^{K_d} \mu_t^{dk}(x)\frac{\mathbb{I}[x_1^d = k] - x}{1-t}. \qquad (17)$$

Intuitively, CatFlow learns a *distribution* over the conditional trajectories to all corners of the probability simplices, rather than regressing towards an expected conditional trajectory.

In the categorical setting, the MF-VFM objective can be written out explicitly. Writing out the probability mass function of the categorical distribution, we see that

$$\log q_t^\theta(x_1^d \mid x) = \log \prod_{k=1}^{K^d}(\mu_t^{dk}(x))^{\mathbb{I}[x_1^d=k]} = \sum_{d=1}^{D}\mathbb{I}[x_1^d = k]\log \mu_t^{dk}(x). \qquad (18)$$

As such, we find that *CatFlow* objective is given by a standard cross-entropy loss:

$$\mathcal{L}_{\text{CatFlow}}(\theta) = -\mathbb{E}_{t,x,x_1}\left[\sum_{d=1}^{D}\sum_{k=1}^{K^d}\mathbb{I}[x_1^d = k]\log \mu_t^{dk}(x)\right]. \qquad (19)$$

Note, however, that when actually computing $v_t^\theta$, this can be done efficiently, since

$$\mathbb{E}_{q_t^\theta(x_1^d|x)}\left[u_t(x^d \mid x_1^d)\right] = \mathbb{E}_{q_t^\theta(x_1^d|x)}\left[\frac{x_1^d - x^d}{1-t}\right] = \frac{\mu^d(x) - x^d}{1-t}, \qquad (20)$$

since $\mu^d(x) := \mathbb{E}_{q_t^\theta(x_1^d|x)}[u_t^d(x_1^d \mid x)]$ and the other terms are not in the expectation. Note that this geometrically corresponds to learning the mapping to a point in the probability simplex, and then flowing towards that. This procedure is illustrated in fig. 1. Because of this, training CatFlow is no less efficient than flow matching. We provided the algorithm for training and generation for CatFlow and a Gaussian VFM objective in appendix B.2 and appendix B.3 respectively.

More precisely, training CatFlow offers two key benefits over standard flow matching. First, given that CatFlow predicts points in the probability simplex and parametrizes the velocities to point into it, an inductive bias is introduced ensuring all generative paths align with realistic trajectories, hence avoiding misaligned paths. Second, using a cross-entropy loss instead of a mean-squared error improves gradient behavior during training. Both aspects enhance learning dynamics and speed up convergence, which is evaluated in section 6.3. Lastly, CatFlow's ability to learn probability vectors – rather than directly choosing classes as is common in discrete approaches – allows the model to express uncertainty about variables at a specific time. This is especially useful in complex domains like molecular generation, where initial uncertainty about components decreases as more structure is established, leading to more precise predictions.

**Permutation Equivariance.** Graphs, defined by vertices and edges, lack a natural vertex order unlike other data types. This permutation invariance means any vertex labeling represents the same graph if the connections remain unchanged. Note that even though the following results apply to graphs, an unordered set of categorical variables can be described by a graph without edges. Under natural conditions – see appendix A – we ensure CatFlow adheres to this symmetry (see appendix A.4 for the proof).

**Theorem 2.** CatFlow generates exchangeable distributions, i.e. CatFlow generates all graph permutations with equal probability.

## 4   Flow Matching and Score Matching: Bridging the Gap

In this section, we relate the VFM framework to existing generative modeling approaches. First, we show that VFM has standard flow matching as a special case when the variational approximation is Gaussian. This implies that VFM provides a more general approach to learning CNFs. Second, we show that through VFM, we are not only able to compute the target vector field, but also the score function as used in score-based diffusion. This has two primary theoretical implications: 1) VFM simultaneously learns deterministic and stochastic dynamics – as diffusion models rely on stochastic dynamics, and 2) VFM provides a variational bound on the model likelihood.

**Relationship to Flow Matching.** VFM admits FM as a special case, under certain assumptions on $u_t(x \mid x_1)$, when the variational approximation is Gaussian. Formally, the following holds (see theorem 3 in appendix A.2 for the proof):

**Theorem 3.** Assume the conditional vector field $u_t(x \mid x_1)$ is linear in $x_1$ and is of the form

$$u_t(x|x_1) = A_t(x)x_1 + b_t(x), \tag{21}$$

where $A_t(x) : [0,1] \times \mathbb{R}^D \to \mathbb{R}^D \times \mathbb{R}^D$ and $b_t(x) : [0,1] \times \mathbb{R}^D \to \mathbb{R}^D$. Moreover, assume that $A_t(x)$ is an invertible matrix and $q_t^\theta(x_1 \mid x) = \mathcal{N}(x_1 \mid \mu_t^\theta(x), \Sigma_t(x))$, where $\Sigma_t(x) = \frac{1}{2}(A_t^\top(x)A_t(x))^{-1}$. Then, VFM reduces to flow matching.

**Relationship to Score-Based Models.** Flow matching [27] is inspired by score-based models [54] and shares strong connections with them. This leads us to two observations. The first is that it should be possible to define a variational parameterization of the score function in diffusion models that is analogous to the one in VFM. The second is that we can build on existing results that show that many diffusion model objectives, including the standard flow matching objective with a linear interpolant, can be expressed as special cases of a general weighted loss function [22, 21]. Here we similarly define a bound on the log-likelihood in terms of a reweighted VFM objective.

In score-based models, the objective is to approximate the score function $\nabla_x \log p_t(x)$ with a function $s_t^\theta(x)$. A connection to VFM becomes apparent by observing that the score function can also be

expressed as an expectation with respect to $p_t(x_1 \mid x)$ (see appendix A.5 for derivation):

$$\nabla_x \log p_t(x) = \int p_t(x_1 \mid x)\nabla_x \log p_t(x \mid x_1)\mathrm{d}x_1 = \mathbb{E}_{p_t(x_1 \mid x)}\left[\nabla_x \log p_t(x \mid x_1)\right], \quad (22)$$

where $\nabla_x \log p_t(x \mid x_1)$ is the tractable conditional score function. Similarly, we can parameterize $s_t^\theta(x)$ in terms of an expectation with respect to a variational approximation $q_t^\theta(x_1 \mid x)$,

$$s_t^\theta(x) := \int q_t^\theta(x_1 \mid x)\nabla_x \log p_t(x \mid x_1) \, \mathrm{d}x_1. \quad (23)$$

It is now clear that $s_t^\theta(x) = \nabla_x \log p_t(x)$ when $q_t^\theta(x_1 \mid x) = p_t(x_1 \mid x)$. This suggests that there exists a variational formulation of score-based models that is entirely analogous to VFM. Indeed, existing work on continuous diffusion for categorical data [11] defines a parameterization of the score function of this form (see section 5 for a more detailed discussion).

Following [54, 3], we can construct stochastic generative dynamics $dx = \tilde{v}_t^\theta(x)dt + g_t dw$ to approximate the true dynamics $dx = \tilde{u}_t(x)dt + g_t dw$ (see details in appendix A.6), with

$$\tilde{u}_t(x) := \mathbb{E}_{p_t(x_1 \mid x)}\left[u_t(x \mid x_1) + \frac{g_t^2}{2}\nabla_x \log p_t(x \mid x_1)\right], \quad \tilde{v}_t^\theta(x) := v_t^\theta(x) + \frac{g_t^2}{2}s_t^\theta(x). \quad (24)$$

Here $g_t : [0, 1] \to \mathbb{R}_+$ is a scalar function, and $w$ is a standard Wiener process.

This connection has two important implications. First, it shows that learning a variational approximation $q_t^\theta(x_1 \mid x)$ can be used to define both deterministic and stochastic dynamics, whereas flow matching typically considers deterministic dynamics only (as flows are viewed through the lens of ODEs). Second, it enables us to show that a reweighted version of the VFM objective provides a bound on the log-likelihood of the model. This result, inspired by [21], provides another theoretical motivation for learning using the VFM objective.

**Theorem 4.** Rewrite the Variational Flow Matching objective as follows:

$$\mathcal{L}_{\text{VFM}}(\theta) = \mathbb{E}_{t,x}\left[\mathcal{L}^\theta(t, x)\right] \quad \text{where} \quad \mathcal{L}^\theta(t, x) = -\mathbb{E}_{x_1}\left[\log q_t^\theta(x_1 \mid x)\right]. \quad (25)$$

Then, the following holds:

$$-\mathbb{E}_{x_1}\left[\log q_1^\theta(x_1)\right] \le \mathbb{E}_{t,x}\left[\lambda_t(x)\mathcal{L}^\theta(t, x)\right] + C, \quad (26)$$

where $\lambda_t(x)$ is a non-negative function and $C$ is a constant.

We provide a proof in appendix A.7. We further note that that applying the same linearity condition that we discussed in section 3.2 to the conditional score function maintains all the same connections with score-based models.

## 5 Related Work

**Diffusion Models for Discrete Data.** Several approaches to diffusion models have been developed for graph generation. In [58], the authors define a Markov process that progressively edits graphs by adding or removing edges and altering node or edge categories and is trained using a graph transformer network, that reverses this process to predict the original graph structure from its noisy version. This approach breaks down the complex task of graph distribution learning into simpler node and edge classification tasks. Moreover, [20] proposes a score-based generative model for graph generation using a system of stochastic differential equations (SDEs). The model effectively captures the complex dependencies between graph nodes and edges by diffusing both node features and adjacency matrices through continuous-time processes. These are non-autoregressive graph generation approaches that perform on par with autoregressive ones, such as in [31, 26, 36]. Other non-autoregressive approaches worth mentioning are [28, 34, 33].

There is also work on diffusion-based models for discrete data in the form of text and other sequential data [4, 33, 61, 18]. Though not explicitly formulated in terms of a variational perspective, the work on continuous diffusion for categorical data [11] arrives at a an approach that is closely related to that of CatFlow. This work defines a diffusion in the embedding space of a transformer-based language

Table 1: Results abstract graph generation.

| | Ego-small | | | Community-small | | |
|---|---|---|---|---|---|---|
| | Degree ↓ | Clustering ↓ | Orbit ↓ | Degree ↓ | Clustering ↓ | Orbit ↓ |
| GraphVAE [50] | 0.130 | 0.170 | 0.050 | 0.350 | 0.980 | 0.540 |
| GNF [29] | 0.030 | 0.100 | 0.001 | 0.200 | 0.200 | 0.110 |
| EDP-GNN [37] | 0.052 | 0.093 | 0.007 | 0.053 | 0.144 | 0.026 |
| GDSS [20] | 0.021 | **0.024** | **0.007** | 0.045 | **0.086** | **0.007** |
| **CatFlow** | **0.013** | **0.024** | 0.008 | **0.018** | **0.086** | **0.007** |

model. It defines an approximation of the score function as an expected value of a conditional score function as in eq. (23), which leads to an expression for the learned score function in terms of a mean embedding, analogous to the one we obtain in eq. (15). Where this approach differs from CatFlow, other than in that it defines a flow in the embedding space of language models, is in how the objective is defined. The authors also minimize a cross-entropy loss, but employ a time-warped objective, similar to the general weighted objective proposed in [22], where the warping is optimized to ensure that the entropy of predictions decreases linearly when moving from noise to data in uniform time.

**Flow-based methods for Discrete Data.** Recently, two flow-based methods for discrete generative modeling have been proposed, which differ both in terms of technical approach and intended use case from the work that we present here.[2]

In [56], a Dirichlet flow framework for DNA sequence design is introduced, utilizing a transport problem defined over the probability simplex, similar to diffusion on simplices proposed in [43]. This approach differs from CatFlow in that it represents the conditional probability path $p_t(x \mid x_1)$ using a Dirichlet distribution. This implies that points $x$ are constrained to the simplex, which is not the case for CatFlow. Dirichlet Flows have not been evaluated on graph generation, but we did carry out preliminary experiments based on the released source code. We compare to this approach in our experiments.

In [6], Discrete Flow Models (DFMs) are introduced. DFMs use Continuous-Time Markov Chains to enable flexible and dynamic sampling in multimodal generative modeling of both continuous and discrete data. Though sharing a goal, this approach differs significantly from CatFlow as in the end the resulting model does not learn a CNF, but rather generation through sequential sampling from a time-dependent categorical distribution. As in the case of Dirichlet flows, no evaluation on graph generation was performed.

The switch to the variational perspective is inspired by [58], showing significant improvement through viewing the dynamics as a classification task over end points. However, CatFlow is still a continuous model, and integrates – rather than iteratively samples – during generation.

## 6 Experiments

We evaluate CatFlow in three sets of experiments. First, we consider an abstract graph generation task proposed in [35], where the goal of this task is to evaluate if CatFlow is able to capture the topological properties of graphs. Second, we consider two common molecular benchmarks, QM9 [40] and ZINC250k [19], consisting of small and (relatively) large molecules respectively. This task is chosen to see if CatFlow can learn semantic information in graph generation, such as molecular properties. Finally, we perform an ablation comparing CatFlow to standard flow matching, specifically in terms of generalization. The experimental setup and model choices are provided in appendix D.

Note that we treat graphs as purely categorical/discrete objects and do not consider 'geometric' graphs that are embedded in e.g. Euclidean space. Specifically, for some graph with $K_v$ node classes and $K_e$ edge classes, we process the graph as a fully-connected graph, where each node is treated as a

---

[2]Flow matching and diffusion models have also been proposed for geometric graph generation, e.g. in [23, 55] and [17, 57] respectively, but since these approaches are continuous (as they generate coordinates based on some conformer) they consider a fundamentally different task than the ones we consider here.

Table 2: Results molecular generation.

| | QM9 | | | ZINC250k | | |
|---|---|---|---|---|---|---|
| | Valid ↑ | Unique ↑ | FCD ↓ | Valid ↑ | Unique ↑ | FCD ↓ |
| MoFlow [63] | 91.36 | 98.65 | 4.467 | 63.11 | 99.99 | 20.931 |
| EDP-GNN [37] | 47.52 | 99.25 | 2.680 | 82.97 | 99.79 | 16.737 |
| GraphEBM [30] | 8.22 | 97.90 | 6.143 | 5.29 | 98.79 | 35.471 |
| GDSS [20] | 95.72 | 98.46 | 2.900 | 97.01 | 99.64 | 14.656 |
| Digress [58] | 99.00 | 96.20 | - | - | - | - |
| Flow Matching [27] | 94.10 | 98.20 | 5.155 | 94.01 | 96.68 | 18.764 |
| Dirichlet FM [56] | 99.10 | 98.15 | 0.888 | 97.52 | 99.20 | 14.222 |
| **CatFlow** | **99.81** | **99.95** | **0.441** | **99.21** | **100.00** | **13.211** |

categorical variable of one of $K_v$ classes and each edge of $K_e + 1$ classes, where the extra class corresponds with being absent.

## 6.1 Abstract graph generation

We first evaluate CatFlow on an abstract graph generation task, including synthetic and real-world graphs. We consider 1) Ego-small (200 graphs), consisting of small ego graphs drawn from a larger Citeseer network dataset [48], 2) Community-small (100 graphs), consisting of randomly generated community graphs, 3) Enzymes (587 graphs), consisting of protein graphs representing tertiary structures of the enzymes from [47], and 4) Grid (100 graphs), consisting of 2D grid graphs. We follow the standard experimental setup popularized by [62] and hence report the maximum mean discrepancy (MMD) to compare the distributions of degree, clustering coefficient, and the number of occurrences of orbits with 4 nodes between generated graphs and a test set. Following [20], we also use the Gaussian Earth Mover's Distance kernel to compute the MMDs instead of the total variation.

The results of the Ego-small and Community-small tasks are summarized in table 1, and additional results (and error bars) are provided in appendix C. The results indicate that CatFlow is able to capture topological properties of graphs, and performs well on abstract graph generation.

## 6.2 Molecular Generation: QM9 & ZINC250k

Molecular generation entails designing novel molecules with specific properties, a complex task hindered by the vast chemical space and long-range dependencies in molecular structures. We evaluate CatFlow on two popular molecular generation benchmarks: QM9 and ZINC250k [40, 19].

We follow the standard setup – e.g. as in [49, 33, 58, 20] – of kekulizing the molecules using RDKit [25] and removing the hydrogen atoms. We sample 10,000 molecules and evaluate them on validity, uniqueness, and Fréchet ChemNet Distance (FCD) – evaluating the distance between data and generated molecules using the activations of ChemNet [39]. Here, validity is computed without valency correction or edge resampling, hence following [63] rather than [49, 33], as is more reasonable due to the existence of formal charges in the data itself. We do not report novelty for QM9 and ZINC250k, as QM9 is an exhaustive list of all small molecules under some chemical constraint and all models obtain (close to) 100% novelty on ZINC250k.[3]

---

[3]CatfFlow obtains 49% novelty on QM9.

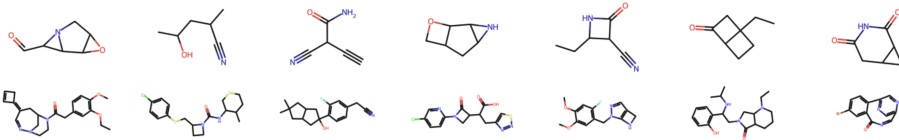

Figure 2: CatFlow samples of QM9 (top) and ZINC250k (bottom).

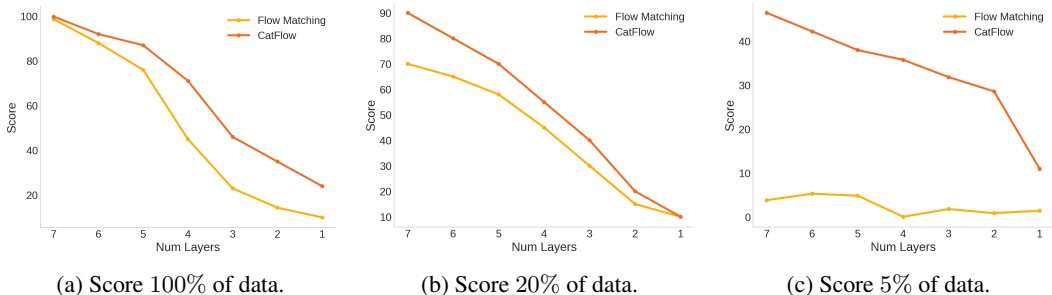

(a) Score 100% of data.    (b) Score 20% of data.    (c) Score 5% of data.

Figure 3: Ablation results. We compare standard flow matching and CatFlow. We visualize performance degradation in terms of a score, which is the percentage of molecules that is valid and unique, for a varying number of layers and percentage of the training data.

The results are summarized in table 2 and samples from the model are shown in fig. 2. CatFlow obtains state-of-the-art results on both QM9 and ZINC250k, virtually obtaining perfect performance on both datasets. It is worth noting that CatFlow also converges faster than flow matching and is not computationally more expensive than any of the baselines either during training or generation.

### 6.3 CatFlow Ablations

To understand the difference in performance between CatFlow and a standard flow matching formulation we perform ablations. we focus on generalization capabilities, and as such consider ablations that the number of parameters in the model and the amount of training data.

In fig. 3 we report a score, which is the percentage of generated molecules that is valid *and* unique. CatFlow not only outperforms regular flow matching in the large model and full data setting, but is also significantly more robust to a decrease in model-size and data. Moreover, we observe significantly faster convergence (curves not shown). We hypothesize this is a consequence of the optimization procedure not exploring 'irrelevant' paths that do not point towards the probability simplex.

## 7 Conclusion

We have introduced a variational reformulation of flow matching. This formulation in turn informed the design of a simple flow matching method for categorical data, which achieves strong performance on graph generation tasks. Variational flow is very general and opens up several lines of inquiry. We see immediate opportunities to apply CatFlow to other discrete data types, including text, source code, and more broadly to the modeling of mixed discrete-continuous data modalities. Additionally, the connections to score-based models that we identify in this paper, suggest a path towards learning both deterministic and stochastic dynamics.

**Limitations.** While the VFM formulation that we identify in this paper has potential in terms of its generality, we have as yet only considered its application to the specific task of categorical graph generation. We leave other use cases of VFM to future work. A limitation of CatFlow, which is shared with related approaches to graph generation, is that reasoning about the set of possible edges has a cost that is quadratic in the number of nodes. As a result CatFlow does not scale well to e.g. large proteins of $10^4$ or more atoms.

**Ethics Statement.** Graph generation in general, and molecular generation specifically, holds great promise for advancing drug discovery and personalized medicine. However, this technology also poses ethical concerns, such as the potential for misuse in creating harmful substances. In terms of technology readiness, this work is not yet at a level where we foresee direct benefits or risks.

**Acknowledgements.** This project was supported by the Bosch Center for Artificial Intelligence.

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

# A    Proofs and Derivations

## A.1    Derivation of the Variational Flow Matching Objective

We derive eq. (9) that states the equivalence of the optimization of the VFM objective and minimization of the KL divergence between the true endpoint distribution $p_t(x_1 \mid x)$ and the variational approximation $q_t^\theta(x_1 \mid x)$. Note that

$$\min_\theta \; \mathbb{E}_{t,x} \left[ \mathrm{KL}\big(p_t(x_1 \mid x) \,||\, q_t^\theta(x_1 \mid x)\big) \right] = \max_\theta \; \mathbb{E}_{t,x,x_1} \left[ \log q_t^\theta(x_1 \mid x) \right], \tag{27}$$

where $t \sim \mathrm{Uniform}(0,1)$, $x \sim p_t(x)$ and $x_1 \sim p_{\mathrm{data}}(x_1)$.

First, we rewrite the KL divergence as combination of entropy and cross-entropy $\mathrm{KL}\,(p \,||\, q) = \mathrm{H}(p,q) - \mathrm{H}(p)$:

$$\mathbb{E}_{t,x} \left[ \mathrm{KL}\big(p_t(x_1 \mid x) \,||\, q_t^\theta(x_1 \mid x)\big) \right] = \mathbb{E}_{t,x} \left[ \mathrm{H}\big(p_t(x_1 \mid x), q_t^\theta(x_1 \mid x)\big) \right] - \tag{28}$$

$$\mathbb{E}_{t,x} \left[ \mathrm{H}\big(p_t(x_1 \mid x)\big) \right]. \tag{29}$$

We observe that the entropy term does not depend on the parameters $\theta$. Consequently we disregard it when optimising the variational distribution $q_t^\theta(x_1 \mid x)$.

Second, we rewrite the cross-entropy term:

$$\mathbb{E}_{t,x} \left[ \mathrm{H}\big(p_t(x_1 \mid x), q_t^\theta(x_1 \mid x)\big) \right] = -\mathbb{E}_{t,x,x_1} \left[ \log q_t^\theta(x_1 \mid x) \right]. \tag{30}$$

Therefore, the second part of eq. (27) corresponds to negative cross-entropy and minimisation corresponds to maximisation of negative cross-entropy.

## A.2    Flow Matching as a Special Case of Variational Flow Matching

**Theorem 3.** Assume the conditional vector field $u_t(x \mid x_1)$ is linear in $x_1$ and is of the form

$$u_t(x|x_1) = A_t(x)x_1 + b_t(x), \tag{21}$$

where $A_t(x) : [0,1] \times \mathbb{R}^D \to \mathbb{R}^D \times \mathbb{R}^D$ and $b_t(x) : [0,1] \times \mathbb{R}^D \to \mathbb{R}^D$. Moreover, assume that $A_t(x)$ is an invertible matrix and $q_t^\theta(x_1 \mid x) = \mathcal{N}(x_1 \mid \mu_t^\theta(x), \Sigma_t(x))$, where $\Sigma_t(x) = \frac{1}{2}(A_t^\top(x) A_t(x))^{-1}$. Then, VFM reduces to flow matching.

*Proof.* Let us substitute the assumed form of $q_t^\theta(x_1 \mid x)$ into the VFM objective:

$$\mathcal{L}_{\mathrm{VFM}}(\theta) = -\mathbb{E}_{t,x,x_1} \left[ \log q_t^\theta(x_1 \mid x) \right] \tag{31}$$

$$= -\mathbb{E}_{t,x,x_1} \left[ \log \left( (2\pi)^{-D/2} |\Sigma_t(x)|^{-1/2} \exp \left( - \left\| A_t(x) \left( x_1 - \mu_t^\theta(x) \right) \right\|_2^2 \right) \right) \right] \tag{32}$$

$$= \mathbb{E}_{t,x,x_1} \left[ \left\| A_t(x) \left( x_1 - \mu_t^\theta(x) \right) \right\|_2^2 \right] + \frac{1}{2}\mathbb{E}_{t,x,x_1} \left[ D \log (2\pi) + \log |\Sigma_t(x)| \right] \tag{33}$$

$$= \mathbb{E}_{t,x,x_1} \left[ \left\| \left( A_t(x) x_1 + b_t(x) \right) - \left( A_t(x) \mu_t^\theta(x) + b_t(x) \right) \right\|_2^2 \right] + C \tag{34}$$

$$= \mathbb{E}_{t,x,x_1} \left[ \left\| u_t(x \mid x_1) - v_t^\theta(x) \right\|_2^2 \right] + C, \tag{35}$$

which is what we wanted to show. $\qquad\square$

## A.3    Decomposition of the Flow

**Theorem 1.** Assume that the conditional vector field $u_t(x \mid x_1)$ is linear in $x_1$. Then, for any distribution $r_t(x_1 \mid x)$ such that the marginal distributions coincide with those of $p_t(x_1 \mid x)$, the corresponding expectations of $u_t(x \mid x_1)$ are equal, i.e.

$$\mathbb{E}_{r_t(x_1|x)} \left[ u_t(x \mid x_1) \right] = \mathbb{E}_{p_t(x_1|x)} \left[ u_t(x \mid x_1) \right]. \tag{12}$$

*Proof.* Applying the linearity condition, we rewrite the conditional vector field $u_t(x|x_1)$ as such:

$$u_t(x \mid x_1) = A_t(x)x_1 + b_t(x), \tag{36}$$

where $A_t(x) : [0,1] \times \mathbb{R}^D \to \mathbb{R}^D \times \mathbb{R}^D$ and $b_t(x) : [0,1] \times \mathbb{R}^D \to \mathbb{R}^D$. Then, we substitute it into equation:

$$\mathbb{E}_{p_t(x_1|x)}[u_t(x \mid x_1)] = \mathbb{E}_{p_t(x_1|x)}[A_t(x)x_1 + b_t(x)] \tag{37}$$

$$= A_t(x)\mathbb{E}_{p_t(x_1|x)}[x_1] + b_t(x). \tag{38}$$

Then, we know, that for any distribution $r$ such that the marginal distributions coincide with those of $p$ the following holds:

$$\mathbb{E}_r[x] = \mathbb{E}_p[x]. \tag{39}$$

Applying this fact to eq. (38) we obtain:

$$\mathbb{E}_{p_t(x_1|x)}[u_t(x \mid x_1)] = A_t(x)\mathbb{E}_{r_t(x_1|x)}[x_1] + b_t(x) \tag{40}$$

$$= \mathbb{E}_{r_t(x_1|x)}[A_t(x)x_1 + b_t(x)] \tag{41}$$

$$= \mathbb{E}_{r_t(x_1|x)}[u_t(x \mid x_1)], \tag{42}$$

which is what we wanted to show. $\qquad\square$

### A.4 CatFlow

Let $\mathcal{G} = (\mathcal{V}, \mathcal{E})$ be a graph with node and edge features given by $\mathbf{H}_n \in \mathbb{R}^{|\mathcal{V}| \times d_n}$ and $\mathbf{H}_e \in \mathbb{R}^{|\mathcal{V}| \times |\mathcal{V}| \times d_e}$ respectively, and let $x$ denote the graphs and its features. Moreover, let $\pi \in S_{|\mathcal{V}|}$ be a permutation and $\mathbf{P}$ its associated permutation matrix, such that the action of the group is defined as

- $\pi \cdot \mathbf{H}_n := \mathbf{P}\mathbf{H}_n$,
- $\pi \cdot \mathbf{H}_e := \mathbf{P}\mathbf{H}_e\mathbf{P}^\top$.

To simplify notation, we will simply write $\pi \cdot x$ to denote the above operation.

**Lemma 1.** If $\mu_t(x)$ is permutation equivariant w.r.t $S_{|\mathcal{V}|}$, then so is $v_t$.

*Proof.* Note that

$$v_t(\pi \cdot x) = \frac{\mu_t(\pi \cdot x) - \pi \cdot x}{1 - t} = \frac{\pi \cdot \mu_t(x) - \pi \cdot x}{1 - t},$$

where the last step follows from permutation equivariance. Moreover, since $\pi$ acts on $x$ through permutation matrices, we can leverage the distributive property of linear operators, i.e. we conclude that

$$\frac{\pi \cdot \mu_t(x) - \pi \cdot x}{1 - t} = \pi \cdot \frac{(\mu_t(x) - x)}{1 - t} = \pi \cdot v_t(x),$$

finishing the proof. $\qquad\square$

**Theorem 5.** Let $p_0$ be an exchangeable distribution – e.g. a standard normal distribution – and $\mu_t(x)$ be permutation equivariant. Then, all permutations of graphs are generated with equal probability.

*Proof.* By result 1, we know that $\mu_t(x)$ being permutation equivariant implies that $v_t(x)$ is permutation equivariant. Moreover, if we let $\Gamma(x) := x + \int_0^1 v_t(x_t)dt$, the for all $\pi \in S_n$ we have that

$$\Gamma(\pi \cdot x) = \pi \cdot x + \int_0^1 v_t(\pi \cdot x_t)dt = \pi \cdot x + \int_0^1 \pi \cdot v_t(x_t)dt = \pi \cdot \Gamma(x),$$

where again the last step follows by basic properties of linear operators. Therefore, since $p_0$ assigns equal density of all permutations of $x$, the resulting distribution $p_1$ preservers this property, which is what we wanted to show. $\qquad\square$

## A.5 Derivation of the Score Function

In this subsection we want to derive the equation that allows us to express the score function $\nabla_x \log p_t(x)$ in terms of the posterior probability path $p_t(x \mid x_1)$. Notice that

$$\nabla_x \log p_t(x) = \frac{1}{p_t(x)} \nabla_x p_t(x) \tag{43}$$

$$= \frac{1}{p_t(x)} \nabla_x \int p_t(x \mid x_1) p(x_1) dx_1 \tag{44}$$

$$= \frac{1}{p_t(x)} \int p(x_1) \nabla_x p_t(x \mid x_1) dx_1 \tag{45}$$

$$= \int \frac{p_t(x \mid x_1) p(x_1)}{p_t(x)} \nabla_x \log p_t(x \mid x_1) dx_1 \tag{46}$$

$$= \int p_t(x_1 \mid x) \nabla_x \log p_t(x \mid x_1) dx_1 \tag{47}$$

$$= \mathbb{E}_{p_t(x_1 \mid x)} \left[ \nabla_x \log p_t(x \mid x_1) \right]. \tag{48}$$

## A.6 Stochastic Dynamics with Variational Flow Matching

In this subsection, we discuss how the VFM framework can be applied to construct stochastic generative dynamics and how it relates to score-based models.

First, let us consider the marginal vector field $u_t(x)$. It provides the deterministic dynamic that can be written as the following ordinary differential equation (ODE):

$$dx = u_t(x) dt. \tag{49}$$

For the vector field $u_t(x)$ we know that – starting from the distribution $p_0(x)$ – it generates some probability path $p_t(x)$. However, as we know from [54, 3], if we have access to the score function $\nabla_x \log p_t(x)$ of distribution $p_t(x)$, we can construct a stochastic differential equation (SDE) that, staring from distribution $p_0(x)$, generates the same probability path $p_t(x)$:

$$dx = \left[ u_t(x) + \frac{g_t^2}{2} \nabla_x \log p_t(x) \right] dt + g_t dw, \tag{50}$$

where $g : \mathbb{R} \to \mathbb{R}_{\geq 0}$ is a scalar function, and $w$ is a standard Wiener process.

Given that

$$u_t(x) = \mathbb{E}_{p_t(x_1 \mid x)} \left[ u_t(x \mid x_1) \right] \quad \text{and} \quad \nabla_x \log p_t(x) = \mathbb{E}_{p_t(x_1 \mid x)} \left[ \nabla_x \log p_t(x \mid x_1) \right], \tag{51}$$

we can rewrite eq. (50) in the following form:

$$dx = \tilde{u}_t(x) dt + g_t dw, \quad \text{where} \tag{52}$$

$$\tilde{u}_t(x) = \mathbb{E}_{p_t(x_1 \mid x)} \left[ \tilde{u}_t(x \mid x_1) \right] \quad \text{and} \quad \tilde{u}_t(x \mid x_1) = u_t(x \mid x_1) + \frac{g_t^2}{2} \nabla_x \log p_t(x \mid x_1). \tag{53}$$

Importantly, the function $g_t$ does not affect the distribution path $p_t(x)$: it only changes the stochasticity of the trajectories. In the extreme case when $g_t \equiv 0$, the SDE in eq. (50) coincides with the ODE in eq. (49).

Moreover, if we construct the stochastic dynamics in this way, we obtain score-based models as a special case. In score-based models [54] the deterministic process that corresponds to probability path $p_t(x)$ has the following form:

$$dx = u_t(x) dt \quad \text{where} \quad u_t(x) = f_t(x) + \frac{g_t^2}{2} \nabla_x \log p_t(x), \tag{54}$$

where $f_t(x)$ is tractable. Substituting this $u_t(x)$ into $eq.$ (50) we obtain:

$$dx = \left[ f_t(x) + g_t^2 \nabla_x \log p_t(x) \right] dt + g_t dw. \tag{55}$$

The SDE in eq. (55) only depends on the score function, so in score-based models, the aim is to learn the score function.

Now, let us note that similarly to vector field $u_t(x)$, the drift term $\tilde{u}_t(x)$ in eq. (52) can be expressed in terms of an end point distribution $p_t(x_1 \mid x)$. Consequently, having a variational approximation of end point distribution $q_t^\theta(x_1 \mid x)$, allows us to construct an approximation of the drift term $\tilde{u}_t(x)$:

$$\tilde{v}_t^\theta(x) = \mathbb{E}_{q_t^\theta(x_1|x)} [\tilde{u}_t(x \mid x_1)] = v_t^\theta(x) + \frac{g_t^2}{2} s_t^\theta(x), \tag{56}$$

$$\text{where} \quad v_t^\theta(x) = \mathbb{E}_{q_t^\theta(x_1|x)} [u_t(x \mid x_1)] \quad \text{and} \quad s_t^\theta(x) = \mathbb{E}_{q_t^\theta(x_1|x)} [\nabla_x \log p_t(x \mid x_1)]. \tag{57}$$

Thus, we may define the following approximated SDE:

$$dx = \tilde{v}_t^\theta(x)dt + g_t dw \quad \text{or} \quad dx = \left[ v_t^\theta(x) + \frac{g_t^2}{2} s_t^\theta(x) \right] dt + g_t dw. \tag{58}$$

Then, eq. (58) is not just simply a new dynamic, it is a family of dynamics that admits deterministic dynamics as a special case when $g_t \equiv 0$. Importantly, the only thing we need to construct the stochastic process in eq. (58) is an approximation of the end point distributions $q_t^\theta(x_1 \mid x)$. Additionally we know that when $p_t(x_1 \mid x) = q_t^\theta(x_1 \mid x)$, $\tilde{u}_t(x) = \tilde{v}_t^\theta(x)$, the processes coincide for all functions $g_t$. Therefore, we can train the model with the same objective as in VFM.

### A.7 Variational Flow Matching as a Variational Bound on the Log-likelihood

In this subsection, we leverage the connections of VFM with stochastic processes to show that a reweighted integral over the point-wise VFM objective defines a bound on the data likelihood in the generative model.

**Theorem 4.** Rewrite the Variational Flow Matching objective as follows:

$$\mathcal{L}_{\text{VFM}}(\theta) = \mathbb{E}_{t,x} [\mathcal{L}^\theta(t, x)] \quad \text{where} \quad \mathcal{L}^\theta(t, x) = -\mathbb{E}_{x_1} [\log q_t^\theta(x_1 \mid x)]. \tag{25}$$

Then, the following holds:

$$-\mathbb{E}_{x_1} [\log q_1^\theta(x_1)] \le \mathbb{E}_{t,x} [\lambda_t(x) \mathcal{L}^\theta(t, x)] + C, \tag{26}$$

where $\lambda_t(x)$ is a non-negative function and $C$ is a constant.

*Proof.* Let us consider the two stochastic processes, that we discussed in appendix A.6:

$$dx = \tilde{u}_t(x)dt + g_t dw, \quad \text{where} \quad dx = \tilde{v}_t^\theta(x)dt + g_t dw. \tag{59}$$

Note that they both start from the same prior distribution $p_0(x)$. The first one, by design, generates probability path $p_t(x)$ and ends up in the data distribution $p_{\text{data}}(x) = p_1(x)$. The second process generates some probability path $q_t^\theta(x)$ that depends on the variational distribution $q_t^\theta(x_1|x)$.

We want to find a variational bound on KL divergence between $p_1(x)$ and $q_1^\theta(x)$. We start by applying the result from [1] (see Lemma 2.22):

$$\text{KL} (p_1(x_1) \| q_1^\theta(x_1)) \le \mathbb{E}_{t,x} \left[ \frac{1}{2g_t^2} \| \tilde{u}_t(x) - \tilde{v}_t^\theta(x) \|_2^2 \right] \tag{60}$$

$$= \mathbb{E}_{t,x} \left[ \frac{1}{2g_t^2} \left\| \int (p_t(x_1 \mid x) - q_t^\theta(x_1 \mid x)) \tilde{u}_t(x|x_1) dx_1 \right\|_2^2 \right] \tag{61}$$

$$\le \mathbb{E}_{t,x} \left[ \frac{1}{2g_t^2} \left( \int \| (p_t(x_1 \mid x) - q_t^\theta(x_1 \mid x)) \tilde{u}_t(x|x_1) \| dx_1 \right)^2 \right] \tag{62}$$

$$\le \mathbb{E}_{t,x} \left[ \frac{1}{2g_t^2} \left( \int |p_t(x_1 \mid x) - q_t^\theta(x_1 \mid x)| \| \tilde{u}_t(x|x_1) \| dx_1 \right)^2 \right]. \tag{63}$$

Now, let us introduce two auxiliary functions:

$$l_t(x) = \sup_{x_1} \|\tilde{u}_t(x|x_1)\| \quad \text{and} \quad \lambda_t(x) = \frac{l_t(x)}{g_t^2}. \tag{64}$$

If we utilise $\lambda_t(x)$ to write down the following bound, we see that

$$\text{KL}\left(p_1(x_1)\|q_1^\theta(x_1)\right) \le \mathbb{E}_{t,x}\left[\frac{\lambda_t(x)}{2}\left(\int \left|p_t(x_1\mid x) - q_t^\theta(x_1\mid x)\right| dx_1\right)_2^2\right]. \tag{65}$$

Next, we can apply Pinsker's inequality, which states that for two probability distributions $p$ and $q$ the following holds:

$$\int |p(x) - q(x)| dx \le 2\text{KL}(p\|q). \tag{66}$$

Applying it to the inner integral, we have:

$$\text{KL}\left(p_1(x_1)\|q_1^\theta(x_1)\right) \le \mathbb{E}_{t,x}\left[\lambda_t(x)\text{KL}\left(p_t(x_1\mid x)\|q_t^\theta(x_1\mid x)\right)\right]. \tag{67}$$

We may rewrite the left part of inequality as a combination of the data entropy and the model's likelihood, where only the likelihood depends on parameters $\theta$:

$$\text{KL}\left(p_1(x_1)\|q_1^\theta(x_1)\right) = -\text{H}\left(p_1(x_1)\right) - \mathbb{E}_{x_1}\left[\log q_1^\theta(x_1)\right]. \tag{68}$$

The right part of inequality can be rewritten as an expectation of entropy that does not depend on any parameters $\theta$ plus a reweighted VFM objective with weighting coefficient $\lambda_t(x)$:

$$\mathbb{E}_{t,x}\left[\lambda_t(x)\text{KL}\left(p_t(x_1\mid x)\|q_t^\theta(x_1\mid x)\right)\right] = -\mathbb{E}_{t,x}\left[\lambda_t(x)\text{H}(p_t(x_1\mid x))\right] \tag{69}$$

$$-\mathbb{E}_{t,x,x_1}\left[\lambda_t(x)q_t^\theta(x_1\mid x)\right] \tag{70}$$

We see that this reweighted version of the VFM objective defines an upper bound on the model likelihood, which was what we wanted to show. $\qquad\square$

### A.8  Stochastic Dynamics under Linearity Conditions

In this subsection, we discuss the connection between VFM and stochastic dynamics under the condition of linearity in $x_1$ of the conditional vector field $u_t(x\mid x_1)$ and conditional score function $\nabla_x \log p_t(x\mid x_1)$.

As we discuss in section 3.2, under the linearity condition, we may express $u_t(x)$ in terms of any distribution of end points $r_t(x_1\mid x)$ if it has the same marginals as $p_t(x_1\mid x)$. In appendix A.5, we demonstrate that the score function $\nabla_x \log p_t(x)$ can also be expressed in terms of end point distributions $p_t(x_1\mid x)$. Therefore, the score function may also be equally expressed in terms of distribution $r_t(x_1\mid x)$ if it has the same marginals as $p_t(x_1\mid x)$. This fact is easy to show in the same way as we present in appendix A.3.

Consequently, under the linearity conditions, the drift term $\tilde{u}_t(x)$ can also be expressed in terms of $r_t(x_1\mid x)$, as it is just a linear combination of the vector field $u_t(x)$ and score function $\nabla_x \log p_t(x)$. Hence, the transition from the distribution $p_(x_1\mid x)$ to some distribution $r_(x_1\mid x)$ does not affect the discussion of connections of stochastic dynamics in appendix A.6.

Furthermore, the transition from distribution $p_t(x_1\mid x)$ to some distribution $r_t(x_1\mid x)$ does not affect connections of the VFM objective with the model likelihood that we discuss in appendix A.7. It is easy to see that in the derivations, we only rely on functions $\tilde{u}_t(x)$ and $\tilde{v}_t^\theta(x)$ (see eq. (60)). However, as we discussed, they are not affected by the transition from $p_t(x_1\mid x)$ to some $r_t(x_1\mid x)$. Therefore, we may repeat all the same derivations for some $r_(x_1\mid x)$ using a factorized distribution.

# B Algorithms

## B.1 General Variational Flow Matching

---
**Algorithm 1** Variational Flow Matching
---

**# Training**
Sample $x_1$ from data and $x_0 \sim p_0(x_0)$
Sample $t \sim \mathcal{U}(0, 1)$
Compute $x_t = tx_1 + (1-t)x_0$
Compute loss $\mathcal{L} = -\log q_t^\theta(x_1 \mid x_t)$
Backpropagate.

**# Generation**
Sample $x_0 \sim p_0(x_0)$
Solve ODE $x_1 = x_0 + \int_{t=0}^{t=1} \frac{\mathbb{E}_{q_t^\theta}[x_1 \mid x_t] - x_t}{1 - t + \varepsilon} \mathrm{d}t$
Return $x_1$

---

## B.2 Categorical Variational Flow Matching (CatFlow)

---
**Algorithm 2** Categorical Variational Flow Matching (CatFlow)
---

*# Assume $q_t(x_1 \mid x_t) = \prod_{d=1}^{D} \mathsf{Cat}(x_1^d \mid \mu_t^d(x_t))$ where $\mu$ is learnable*

**# Training**
Sample $x_1$ from data and $x_0 \sim p_0(x_0)$
Sample $t \sim \mathcal{U}(0, 1)$
Compute $x_t = tx_1 + (1-t)x_0$
Compute cross-entropy loss $\mathcal{L} = -\sum_{d=1}^{D} \sum_{k=1}^{K^d} \mathbb{I}[x_1^d = k] \log \mu_t^{dk}(x)$
Backpropagate.

**# Generation**
Sample $x_0 \sim p_0(x_0)$
Solve ODE $x_1 = x_0 + \int_{t=0}^{t=1} \frac{\mu_t(x_t) - x_t}{1 - t + \varepsilon} \mathrm{d}t$
Return $x_1$

---

## B.3 Gaussian Variational Flow Matching

---
**Algorithm 3** Gaussian Variational Flow Matching
---

*# Assume $q_t(x_1 \mid x_t) = \prod_{d=1}^{D} \mathcal{N}(x_1^d \mid \mu_t^d(x_t), I)$ where $\mu$ is learnable*

**# Training**
Sample $x_1$ from data and $x_0 \sim p_0(x_0)$
Sample $t \sim \mathcal{U}(0, 1)$
Compute $x_t = tx_1 + (1-t)x_0$
Compute mean squared error loss $\mathcal{L} = \frac{1}{2} \sum_{d=1}^{D} ||\mu_t^d(x_t) - x_1^d||^2$
Backpropagate.

**# Generation**
Sample $x_0 \sim p_0(x_0)$
Solve ODE $x_1 = x_0 + \int_{t=0}^{t=1} \frac{\mu_t(x_t) - x_t}{1 - t + \varepsilon} \mathrm{d}t$
Return $x_1$

---

# C  Additional Results

## C.1  Detailed results

Here, we provide the results for CatFlow with standard deviations, as computed as in [20] through multiple seeds.

| Ego-small | | | Community-small | | |
|---|---|---|---|---|---|
| $4 \leq |\mathcal{V}| \leq 18$ | | | $12 \leq |\mathcal{V}| \leq 20$ | | |
| Degree ↓ | Clustering ↓ | Orbit ↓ | Degree ↓ | Clustering ↓ | Orbit ↓ |
| $0.013 \pm 0.007$ | $0.024 \pm 0.009$ | $0.008 \pm 0.005$ | $0.018 \pm 0.012$ | $0.086 \pm 0.021$ | $0.007 \pm 0.005$ |

| Enzymes | | | Grid | | |
|---|---|---|---|---|---|
| $10 \leq |\mathcal{V}| \leq 125$ | | | $100 \leq |\mathcal{V}| \leq 400$ | | |
| Degree ↓ | Clustering ↓ | Orbit ↓ | Degree ↓ | Clustering ↓ | Orbit ↓ |
| $0.013 \pm 0.012$ | $0.062 \pm 0.011$ | $0.008 \pm 0.007$ | $0.115 \pm 0.010$ | $0.004 \pm 0.002$ | $0.075 \pm 0.071$ |

| QM9 | | | ZINC250k | | |
|---|---|---|---|---|---|
| $1 \leq |\mathcal{V}| \leq 9$, 4 atom types | | | $6 \leq |\mathcal{V}| \leq 38$, 9 atom types | | |
| Valid ↑ | Unique ↑ | FCD ↓ | Valid ↑ | Unique ↑ | FCD ↓ |
| $99.81 \pm 0.03$ | $99.95 \pm 0.02$ | $0.441 \pm 0.023$ | $99.21 \pm 0.04$ | $100.00 \pm 0.00$ | $13.211 \pm 0.12$ |

## C.2  Extra results SBM and Planar Graphs

Results on Stochastic Block Model and Planar Graphs. We ran extra experiments for (standard) flow matching and Dirichlet flow matching. We observe that CatFlow obtains SOTA performance on all tasks and metrics. Moreover, it is worth noting CatFlow was significantly faster to train than Dirichlet FM due to a computationally cheaper forward process.

| Model | Deg ↓ | Clus ↓ | Orb ↓ | V.U.N. ↑ |
|---|---|---|---|---|
| *Stochastic Block Model* | | | | |
| SPECTRE [35] | 1.9 | 1.6 | 1.6 | 53% |
| ConGress [58] | 34.1 | 3.1 | 4.5 | 0% |
| DiGress [58] | 1.6 | **1.5** | 1.7 | 74% |
| Flow Matching [27] | 10.2 | 2.0 | 3.2 | 22% |
| Dirichlet FM [56] | 1.7 | **1.5** | **1.4** | 80% |
| **CatFlow** | **1.5** | **1.5** | **1.4** | **85%** |
| *Planar Graphs* | | | | |
| SPECTRE [35] | 2.5 | 2.5 | 2.4 | 25% |
| ConGress [58] | 23.8 | 8.8 | 2590 | 0% |
| DiGress [58] | **1.4** | **1.2** | 1.7 | 75% |
| Flow Matching [27] | 5.1 | 5.6 | 5.5 | 30% |
| Dirchlet FM [56] | 1.5 | 1.3 | **1.5** | **80%** |
| **CatFlow** | **1.4** | 1.3 | **1.5** | 80% |

# D  Experimental setup

## D.1  Model

To ensure a comparison on equal terms to baselines, we employ the same graph transformer network as proposed in [12], which was also used in [58], along with the same hyper-parameter setup. We summarize the parametrization of our network here.

Just as done in DiGress, our graph transformer takes as input a graph $(\mathbf{H}_n, \mathbf{H}_e)$ and predicts a distribution over the clean graphs, using structural and spectral features to improve the network expressivity, which we denote as $\mathbf{H}_g$. Each transformer layer does the following operations:

1. **Node Features $\mathbf{H}_n$ and Edge Features $\mathbf{H}_e$:**

    (a) **Linear Transformation**: Apply linear transformations to both $\mathbf{H}_n$ and $\mathbf{H}_e$.

    (b) **Outer Product and Scaling**: Compute the outer product of the transformed features and apply scaling.

2. **Node Features $\mathbf{H}_n$:**

    (a) **Feature-wise Linear Modulation (FiLM)**: Apply FiLM to the transformed node features using global features $\mathbf{H}_g$.

3. **Edge Features $\mathbf{H}_e$:**

    (a) **Feature-wise Linear Modulation (FiLM)**: Apply FiLM to the transformed edge features using global features $\mathbf{H}_g$.

4. **Self-Attention Mechanism**:

    (a) **Linear Transformation**: Apply a linear transformation to the transformed node features.

    (b) **Softmax Operation**: Compute the attention scores using the softmax function.

    (c) **Attention Score Calculation**: Calculate the weighted sum of the transformed node features based on the attention scores.

5. **Global Features $y$:**

    (a) **Pooling**: Apply PNA pooling to the node features $\mathbf{H}_n$ and edge features $\mathbf{H}_e$.

    (b) **Summation**: Sum the pooled features with the global features $\mathbf{H}_g$.

6. **Final Outputs**:

    (a) **Node Features $\mathbf{H}_n'$**: Obtain updated node features after the attention mechanism.

    (b) **Edge Features $\mathbf{H}_e'$**: Obtain updated edge features after the attention mechanism.

    (c) **Global Features $\mathbf{H}_g'$**: Obtain updated global features after summation.

Here, the FiLM operation is defined as:

$$\text{FiLM}(M_1, M_2) = M_1 W_1 + (M_1 W_2) \odot M_2 + M_2$$

for learnable weight matrices $W_1$ and $W_2$, and PNA is defined as:

$$\text{PNA}(X) = \text{cat}(\max(X), \min(X), \text{mean}(X), \text{std}(X))W.$$

## D.2  Hyperparameters and Computational Costs

We report the hyperparameters here:

All models were trained until convergence. Furthermore, all data splits are kept the same as in [20], and hidden dimensions are kept the same as [58]. All experiments were run on a single NVIDIA RTX 6000 and took about a day to run.

| Hyperparameter | Abstract | QM9/ZINC250k | Ablation |
|---|---|---|---|
| Optimizer | AdamW | AdamW | AdamW |
| Scheduler | Cosine Annealing | Cosine Annealing | Cosine Annealing |
| Learning Rate | $2 \cdot 10^{-4}$ | $2 \cdot 10^{-4}$ | $2 \cdot 10^{-4}$ |
| Weight Decay | $1 \cdot 10^{-12}$ | $1 \cdot 10^{-12}$ | $1 \cdot 10^{-12}$ |
| EMA | 0.999 | 0.999 | 0.999 |

Table 3: Hyperparameter setup.

## E  Detail CNFs

To compute the resulting distribution $p_t$ for CNF, one can use the change of variables formula:

$$[\varphi_t]_* p_0(x) = p_0(\varphi_t^{-1}(x)) \det \left[ \frac{\partial \varphi_t^{-1}}{\partial x}(x) \right]. \tag{71}$$

This induces a *probability path*, i.e. a mapping $p_t : [0,1] \times \mathbb{R}^D \to \mathbb{R}_{>0}$ such that $\int p_t(x) dx = 1$ for all $t \in [0,1]$. We say that $v_t$ *generates* this probability path given a starting distribution $p_0$. In theory, one could try and optimize the empirical divergence between the resulting distribution $p_1$ and target distribution, but obtaining a gradient sample for the loss requires us to solve the ODE at each step during training, making this approach computationally prohibitive.

One way to assess if a vector field generates a specific probability path is using the continuity equation, i.e. we can assess whether $v_t$ and $p_t$ satisfy

$$\frac{\partial}{\partial t} p_t(x) + \nabla \cdot (p_t(x) v_t(x)) = 0, \tag{72}$$

where $\nabla$ is the divergence operator. Note that by sampling $x_0 \sim p_0$, a new sample from $p_1$ can be generated by following this ODE, i.e. integrating

$$x_1 = x_0 + \int_0^1 v_t(x_t) \mathrm{d}t. \tag{73}$$

