# OpenReview forum: "Variational Flow Matching for Graph Generation"
_NeurIPS.cc/2024/Conference — NeurIPS 2024 poster_

### Official Review · Reviewer_vH9B · 2024-06-26

**Soundness:** 3
**Presentation:** 3
**Contribution:** 2
**Rating:** 5
**Confidence:** 4

**Summary:**

This paper proposed a variant of flow matching for graph generation from a variational perspective, called CatFlow. The main idea is not to estimate the marginal velocity, but instead to estimate the posterior first and then evaluate the marginal velocity as the expectation of the conditional velocity w.r.t. the posterior. In the case of linear conditional velocity and Gaussian variational approximation, the equivalence to vanilla flow matching has been established and connection to score-based models have been discussed. A simplified mean-field variational approach has also been proposed based on the observation that marginal posterior matching (for individual dimensioins) is enough for posterior mean estimation for linear conditional velocity. Experiments on several graph/molecule generation tasks demonstrate the effectiveness of the CatFlow.

**Strengths:**

The paper is written clearly. The variational formulation and the connection to standard flow matching is new. The authors have also shown that a weighted average of the training objective of VFM provides a bound on the log-likelihood of the model.

**Weaknesses:**

The main weakness of the paper is that the core idea of learning the posterior first (i.e., the variational formulation) and evaluating the marginal velocity using the expected conditional velocity w.r.t. the posterior has already been proposed in Dirichlet Flow [1]. Although the author claimed that Dirichlet Flow uses the specific dirichlet distribution as the conditonal probability, this is mainly a design choice which does not affect the generality of Dirichlet Flow for modeling discrete objectives such as graphs and molecules. Given this, the lack of an appropriate comparison to Dirichlet Flow in the experiments represents a significant drawback of the current paper.

The paper also lacks a brief introduction to the baseline methods, which would help contextualize the results and better highlight the contributions of the proposed approach.


[1] Hannes Stark, Bowen Jing, Chenyu Wang, Gabriele Corso, Bonnie Berger, Regina Barzilay, and Tommi Jaakkola. Dirichlet flow matching with applications to dna sequence design. arXiv preprint arXiv:2402.05841, 2024.

**Questions:**

1. Although it is reasonable to match the marginal posterior using mean-field approximation, it is still not clear if this simple mean-field approximation is good enough to capture the posterior mean. Can the author provide some intuition when the mean-field approximation is good enough?

2. In flow matching, an error bound for the generated distribution in term of flow matching error is provided. Can a similar result be provided here in terms of the variational approximation error?

3. How would VFM perform when the conditional velocity is not linear?

**Limitations:**

The limitations has been adequately addressed.

---

> ### Author Rebuttal · Authors · 2024-08-07
>
> Dear reviewer vH9B,
>
> We would like to thank you for the useful questions and positive words about our work. We will first answer each question and then propose concrete changes to the paper to address them in the final version.
>
> > The main weakness of the paper is that the core idea of learning the posterior first (i.e., the variational formulation) and evaluating the marginal velocity using the expected conditional velocity w.r.t. the posterior has already been proposed in Dirichlet Flow [1]. Although the author claimed that Dirichlet Flow uses the specific dirichlet distribution as the conditonal probability, this is mainly a design choice which does not affect the generality of Dirichlet Flow for modeling discrete objectives such as graphs and molecules. Given this, the lack of an appropriate comparison to Dirichlet Flow in the experiments represents a significant drawback of the current paper.
>
> We agree that this work is directly related to the recently proposed work on Dirichlet FM and that an experimental comparison is warranted in this context. We have included such a comparison in our main response.
>
> To briefly comment further on the relationship to Dirichlet FM: the cross-entropy loss and the velocity field in equations (9) and (10) of the Dirichlet FM paper are analogous to those in our paper, which (incidentally) are *also* described in Equations (9) and (10) in our manuscript. From a methodological perspective, the main distinction is that the Dirichlet FM paper is framed in terms of a probability path on the simplex, whereas we arrive at CatFlow by way of a more general variational perspective that admits both categorical and continuous models as special cases. Moreover, CatFlow is not defined through a forward process on the simplex and simply operates in $\mathbb{R}^n$.
>
> **Concrete Steps:** As shown in the main response, we have now performed additional experiments to compare against Dirichlet FM. As we wrote, we did make an earlier attempt to perform a comparison at the time of submission, but we felt that the (poor) performance we observed in our initial results was not sufficiently representative to merit inclusion. We believe that we have addressed this with these new results and hope this also addresses the main concern raised by the reviewer.
>
> > The paper also lacks a brief introduction to the baseline methods, which would help contextualize the results and better highlight the contributions of the proposed approach.
>
> That is a fair point!
>
> **Concrete steps:** We will mention the most important baselines in the main text, and add a small description of the other ones in the appendix.
>
> > Although it is reasonable to match the marginal posterior using mean-field approximation, it is still not clear if this simple mean-field approximation is good enough to capture the posterior mean. Can the author provide some intuition when the mean-field approximation is good enough?
>
> We are not sure why this might be a concern, perhaps the reviewer could elaborate?
>
> Mean-field variational inference is known to under-approximate the posterior variance when minimizing an exclusive/reverse KL divergence. This is a consequence of the fact that the variational family cannot capture correlations in the posterior. However, the posterior mean estimates are generally known to be accurate. As such, we have *no* indication that these distributions would *not* be able to provide a good estimate of the mean. This also appears to be born out by empirical results.
>
> **Concrete Steps:** We will amend the paper to provide some discussion on this point.
>
> > In flow matching, an error bound for the generated distribution in term of flow matching error is provided. Can a similar result be provided here in terms of the variational approximation error?
>
> **Answer:**  We are not aware of error bounds in the concurrent works on Flow Matching by Lipman et al.,  Albergo at al., and Liu et al. Is the reviewer referring to the error bounds developed by Benton and colleagues [1]? It might indeed be possible to derive an analogous result starting from an assumed bound on the approximation error in the posterior mean, rather than an approximation error in the velocity field. However, this is not something that we have considered in the context of this submission.
>
> [1] Joe Benton, George Deligiannidis, Arnaud Doucet. Error Bounds for Flow Matching Methods. TMLR 2024.
>
> > How would VFM perform when the conditional velocity is not linear?
>
> This is a very interesting question.
>
> Just to make sure we are on the same page: The linearity assumption we care about is *the linearity of the conditional field in the endpoint*, i.e. the fact that $u_t(x \mid x_1)$ is linear in $x_1$. This is subtly different from the fact that in flow matching, $u_t(x \mid x_1)$ itself is a linear interpolation, which is a design choice in FM typically referred to as the ‘optimal transport’ formulation.
>
> The linearity in $x_1$ is, however, a standard assumption often made in e.g. diffusion models and flow matching models. We show that under this common linearity assumption, optimising VFM using a 'mean-field approximation' can actually find the exact solution. In this setting the mean-field approximation is not an approximation at all!
>
> The question of what happens if this linearity assumption does not hold is one which, as far as we know, cannot be tackled by the standard diffusion/flow matching techniques out there. However, using VFM one could simply resolve to learning the loss $\mathcal{L}(\theta) = -\mathbb{E}[\log q_t^\theta (x_1 \mid x)]$ through stochastic gradients for some distribution $q_t^\theta$ that is not fully factorized. This is actually a promising direction for future research and might be a good example of where a variational point of view can offer new perspectives relative to the standard point of view.
>
> Thank you so much for your time and effort reviewing our work.

---

> > ### Author Response · Authors · 2024-08-12
> >
> > Dear reviewer vH9B, we wondered if the added results and comparison to Dirichlet FM addressed your main concerns regarding our work. Thanks a lot for your review once more.

---

### Official Review · Reviewer_WAZn · 2024-07-13

**Soundness:** 3
**Presentation:** 3
**Contribution:** 3
**Rating:** 6
**Confidence:** 3

**Summary:**

This work presents variational flow matching framework VFM and introduces CatFlow, an application to generation of categorical data such as graph. The paper reformulates flow matching using variational perspective and with linear assumption on the conditional vector field achieves tractable objective. The authors derive the connection with the score-based models and show that the framework provides variational  bound on the model likelihood.

**Strengths:**

- To the best of my knowledge, the variational framework for flow matching is novel and the provided theoretical works seem solid.
- Connection with the score-based model provide intuitions on the method.
- Application to discrete data seems reasonable and the benefits over standard flow matching is well explained.

**Weaknesses:**

- The reason CatFlow outperforms previous diffusion models is unclear. Is it because VFM provides variational bound on the model likelihood? Ablation studies on why CatFlow outperforms other diffusion models would strengthen the work.
- CatFlow should be validated on larger datasets used in recent works [1, 2], for example Planar and SBM, since community-small and ego-small datasets consists of very small graphs which is not suitable for evaluating generative models.
- In particular, validity metric should be used to evaluate the method, instead of only relying on MMD. Evaluating only with MMD is not appropriate as MMD may fail to catch important graph characteristics, for example, small MMD does not guarantee the generated graphs is actually a community graph. Evaluating on Planar or SBM dataset and comparing the results of V.U.N. (valid, unique, and novel) is necessary to strongly argue the advantage of the proposed method.
- Comparison with discrete diffusion model like DiGress should be done on more datasets other than QM9. This also leads to experiments on Planar or SBM datasets or in molecular generation tasks, like GuacaMol dataset.

Minor correction: citation in line 148 should be fixed

**Questions:**

I would appreciate it if the authors could address the weakness above.
Also, I recommend more explanation on Figure 1 as it would privde the readers to get better understanding on CatFlow.

In summary, I believe the paper provides theoretical contributions, but the experimental validation needs more room to improve.

**Limitations:**

The authors have addressed the limitations at the end of the paper.

---

> ### Author Rebuttal · Authors · 2024-08-07
>
> Dear reviewer WAZn,
>
> We would like to thank you very much for your positive comments and thoughtful questions about our work. We will first answer each question and then propose concrete changes to the paper to address them in the final version. Since some questions are related, we grouped them together in our answer.
>
> > The reason CatFlow outperforms previous diffusion models is unclear. Is it because VFM provides variational bound on the model likelihood? Ablation studies on why CatFlow outperforms other diffusion models would strengthen the work.
> >
>
> Thanks a lot for this question. Our hypothesis is that CatFlow outperforms diffusion-based baselines for similar reasons that continuous FM has proven a strong competitor to diffusion in e.g. image generation, which is to say that a linear interpolation in the probability path appears to work well in practice.  It is of course difficult to demonstrate conclusively why any method works better than any other method. Indeed, to our knowledge no such demonstration exists for continuous FM either.
>
> **Concrete steps:** We will discuss the intuition for why CatFlow might outperform diffusion-based baselines  more clearly in the text.
>
> > CatFlow should be validated on larger datasets used in recent works [1, 2], for example Planar and SBM, since community-small and ego-small datasets consists of very small graphs which is not suitable for evaluating generative models.
> In particular, validity metric should be used to evaluate the method, instead of only relying on MMD. Evaluating only with MMD is not appropriate as MMD may fail to catch important graph characteristics, for example, small MMD does not guarantee the generated graphs is actually a community graph. Evaluating on Planar or SBM dataset and comparing the results of V.U.N. (valid, unique, and novel) is necessary to strongly argue the advantage of the proposed method.
> Comparison with discrete diffusion model like DiGress should be done on more datasets other than QM9. This also leads to experiments on Planar or SBM datasets or in molecular generation tasks, like GuacaMol dataset.
> >
>
> Thank you for the suggestion – we fully agree! We have run CatFlow on Planar and SBM now and have obtained SOTA performance on these datasets as well, being on par / a tiny bit better than Digress on all metrics, including V.U.N. (all reported in the added PDF). We will report these values in the final version of the paper. Moreover, we would like to highlight that in the submitted manuscript, we also ran large-graph experiments (see appendix B), on which we did obtain strong performance as well.
> Sadly, we have not been able to run the GuacaMol dataset in time for the response deadline, but we will aim to include that in the final version of the paper.
>
> **Concrete steps:** Add results Planar and SBM in the final version of the paper, including extra metrics. Where appropriate, we will rerun the existing experiments with more metrics to provide a better comparison to existing methods.
>
> We will also make sure the citation mentioned will be corrected and the text in the figure will be improved.
>
> Last, thank you so much for your time reviewing our work.

---

### Official Review · Reviewer_5EeZ · 2024-07-16

**Soundness:** 4
**Presentation:** 4
**Contribution:** 4
**Rating:** 7
**Confidence:** 3

**Summary:**

This paper introduces a new variational inference framework of flow matching with a focus on applying the framework on discrete data generation. Instead of using the squared norm in standard flow matching, the paper proposes a variational distribution to the conditional path, which is used in the vector field. The paper shows that when the variational distribution is identical to the conditional path, the approximate vector field equals to the target one. The paper then uses mean-field factorisation of the variational distribution and applies linear conditional vector field. Based on the proposed framework, the paper mainly applies it for discrete data generation including the tasks of abstract graph generation and molecular generation. Experiments show that the proposed method outperforms other diffusion-based or flow-matching-based methods.

**Strengths:**

1. The technical contribution and significance of the paper are good. The paper provides a novel angle of formulating flow matching as a variational inference problem, which is theoretically sound and conceptually intuitive.

2. The paper derives a general framework first and then implement it on the task of discrete data generation. The paper also provides comprehensive theoretical analysis on the connections to related methods.

3. The experimental results of the proposed method are convincing.

4. The paper is well-written in general.

**Weaknesses:**

In general, the quality of the paper is good. But I have a few questions on clarity. Please see below.

**Questions:**

but not all of them are weaknesses.

1. In standard flow matching, one needs to minimise the squared norm between the parameterised vector field and the true vector field. That's why one needs to compute the (conditional) vector field. But in variational flow matching, it seems that the final objective is the maximisation of the data likelihood in terms of $\theta$. I wonder how the vector fields come into play in this case. Are they used for computing the variational distribution somehow?

2. Related to the previous question, it would be good to have more details of the parameterisation of $q$ or the implementation of $\theta$.

3. Mean-field approximation of the variational distribution and linear formulation of the conditional vector field are used, more for efficiency consideration.  Can the authors provide more discussions on when these approximation and assumption will be the bottleneck of performance?

4. Although the code is provided, it might be good to provide an algorithm of pseudo code in the paper.

5. It would be good to have more explanations of Figure 1.

**Limitations:**

The authors adequately addressed the limitations

---

> ### Author Rebuttal · Authors · 2024-08-07
>
> Dear reviewer 5EeZ,
>
> First of all, we would like to thank you very much for your positive words and useful points about clarity. We will first answer each question and then propose concrete changes to the paper to address them in the final version.
>
> > In standard flow matching, one needs to minimise the squared norm between the parameterised vector field and the true vector field. That's why one needs to compute the (conditional) vector field. But in variational flow matching, it seems that the final objective is the maximisation of the data likelihood in terms of $\theta$. I wonder how the vector fields come into play in this case. Are they used for computing the variational distribution somehow?
>
>
> If we understand the question correctly, the reviewer is wondering how the role of the conditional vector field in VFM differs from its role in standard FM, particularly during training.
>
> During *generation*, we explicitly use $u_t(x | x_1)$ to compute the approximate velocity field  $v_t^{\theta}(x) = \mathbb{E}_{q_t}[u_t(x | x_1)]$.
> However, during *training*, we indeed do *not* compute it explicitly to evaluate the objective.
>
> With that said, the conditional velocity field $u_t(x | x_1)$ generates a probability path $p_t(x | x_1)$, which in turn defines a posterior probability path $p_t(x_1 | x)$, which we approximate with a variational distribution $q_\theta(x_1 | x)$. In other words, the learned variational approximation implicitly depends on the choice of conditional velocity field $u_t(x | x_1)$, even if this field does not show up explicitly in the the objective. Please let us know if this answers your question, we would be happy to clarify further.
>
> **Concrete steps:** We will dedicate a paragraph in the paper to explaining this better. We will also add one algorithm block for training and one for sampling/generation that clearly summarises the steps.
>
> > Related to the previous question, it would be good to have more details of the parameterisation of $q$ or the implementation of $\theta$.
> >
>
> Yes, we fully agree! Since the goal in VFM is to predict $\mathbb{E}[x_1 \mid x_t]$, the goal is to train a network to predict the parameter needed to compute this expectation based on input $x_t$ at time $t$, e.g. the mean of a Gaussian distribution or the probability vector of a categorical distribution.
>
> **Concrete steps:** We will add a detailed section in the appendix describing the parameterisation of $q$ and $\theta$. Specifically, we will do this for 1) the general case, 2) the Gaussian case, and 3) the Categorical case in the final version.
>
> > Mean-field approximation of the variational distribution and linear formulation of the conditional vector field are used, more for efficiency consideration. Can the authors provide more discussions on when these approximation and assumption will be the bottleneck of performance?
> >
>
> As we understand it, the reviewer has two related questions here,  pertaining how model expressivity/performance are affected by (1) the  mean-field parameterization and (2) the assumption of linearity of the conditional vector field in $x_1$.
>
> *Implications of (1)*: Theorem 1  states that if (2) holds, the mean-field parameterization is not an “approximation’’ as much as it is a “simplification”.
>
> The approximate flow field $v_\theta(x)$ matches the flow field $u(x)$ *exactly* whenever for each component of the posterior mean $\mu_1^d$ under $p_t$ we have that $\mu_1^d =  \mathbb{E}_{q_t}[x_1^d | x]$.
> This is to say that the *only* requirement in VFM under assumption (2) is that each component $d$ of the mean of the variational distribution must match the mean of the posterior probability path. A mean-field parameterization therefore does not compromising expressivity/performance at all in this setting.
>
> *Implications of (2)*: The assumption of linearity in $x_1$ for the conditional flow field holds for most existing flow matching methods and indeed an analogous assumption hold for many diffusion-based models as well. We would therefore not consider this a particularly strong restriction in practice, and certainly not something that might become a bottleneck for model performance relative to existing flow matching and diffusion-based methods.
>
> **Concrete steps:** We will dedicate an extra paragraph in the paper to elaborating on these points.
>
> > Although the code is provided, it might be good to provide an algorithm of pseudo code in the paper.
> >
>
> We fully agree. This would also help highlight the fact that, from an implementation point of view, VFM is not more complicated than FM.
>
> **Concrete steps:** We will add 1) a general code block, 2) a code block for the Gaussian case, and 3) a code block for the categorical case. We will also add a jupyter notebook with these cases as examples, and we are currently working on a small pytorch library to supply all code which is to be released soon.
>
> > It would be good to have more explanations of Figure 1.
> >
>
> Yes, we agree. Moreover, we think that in general some extra figures should added, with the aim of making the difference between VFM and standard FM more clear.
>
> **Concrete steps:** We will add this description in. Moreover, we will add the extra figures mentioned above.
>
> Again, thank you so much for the time you spent reviewing our work.

---

> > ### Comment · Reviewer_5EeZ · 2024-08-14
> >
> > Thanks for the response. I am happy to keep my original rating of the paper.

---

### Author Rebuttal · Authors · 2024-08-07

We would like to thank the reviewers for their thoughtful reading of the manuscript and their detailed comments. We are happy to hear that reviewers overall appear in agreement that this is a clearly written paper that provides a novel variational perspective on flow matching, develops  useful connections with related approaches, and demonstrates good empirical results on graph generation tasks.

We respond to individual points and questions by each reviewer below. We would like to highlight two points raised by reviewers about additional experimental evaluations. We include results for these evaluations in the attached PDF, and will incorporate these results in the manuscript:

- Reviewer `WAZn` made the helpful suggestion to evaluate on the **Planar** and **SBM** datasets, which we have done. Results show that CatFlow attains SOTA performance on these datasets. We would also like to call attention to additional results on larger graphs in **Appendix B** (which were already in the original submission).
- Reviewer `vH9B` comments that a comparison with **Dirichlet FM** would be helpful. We completely agree. As we wrote in our manuscript, we were unfortunately not able to get this method working sufficiently well in our initial (limited) experiments. We have since invested additional time and effort and are now able to report a comparison that we believe is representative. The results show that even though Dirichlet FM outperforms standard FM in graph generation tasks, CatFlow obtains better performance in the tasks considered in our work. Though not visible in this table, we note that CatFlow also was faster to train in our experiments.
- We added an additional comparison with a baseline trained with the normal **Flow Matching** objective. To adapt this model to categorical data, we simply select the nearest one-hot vector on the simplex at the final step of generation.

Additional changes that reviewers can expect mainly pertain to points of clarification. In each of our responses, we use the text “**Concrete Steps”** to summarise what changes we intend to make in response to reviewer comments.

---

### Public Comment · ~Zijing_Ou1 · 2024-11-28

Hi, thanks for the great work.

I am confused about the generation step in algorithm 2: $x_1 = x_0 + \int\_{t=0}^{t=1} \frac{\mu_t (x_t) - x_t }{1-t+\epsilon} dt$. Did you re-normalize the logits and obtain $x_t$ by sampling from the categorical distribution? How do you ensure that $x_t$ remains a categorical variable when solving this ODE?

---

> ### Public Comment · ~Floor_Eijkelboom1 · 2024-11-28
>
> Dear Zijing,
>
> Thanks a lot for the kind words and reaching out!
>
> That's a good point - we actually do not ensure that $x_t$ is a categorical variable at all times $t$, e.g. $x_0$ is sampled from $x_0 \sim \mathcal{N}(x_0 \mid 0, {I})$ and hence is just a vector in $\mathbb{R}^K$. However, since $\mu^{\theta}(x_t)$ always lies on $\Delta^K$ (as it is the expected value $\mathbb{E}_{q^{\theta}_t}[x_1 \mid x_t]$ for a categorical distribution $q_t^{\theta}(x_1 \mid x_t)$),  we will have that $v^{\theta}_t$ will always points *into* $\Delta^K$. Hence, as $t \to 1$, our $x_t$ will move to a point on $\Delta^K$. Then, to 'select' the class after generation, we simply choose the point with the highest probability / closest corner on the simplex.
>
> I hope this clears up the confusing. If not, feel free to reach out directly by email!

---

> > ### Public Comment · ~Zijing_Ou1 · 2024-11-28
> >
> > Thank you for the clarification! It is clear to me now. I really enjoyed reading the paper—great work once again! 😊

---

### Decision · Program_Chairs · 2024-09-25

**Decision:**

Accept (poster)

**Comment:**

The work suggests the use of variational framework in flow matching, a natural step in an interesting topic. This work is timely and important to the community and will be an excellent addition to NeurIPS.